# A retrospective on hydrological catchment modelling based on half a century with the HBV model

Jan Seibert[1,2] and Sten Bergström[3]

[1] Department of Geography, University of Zurich, Zurich, 8057, Switzerland

[2] Department of Aquatic Sciences and Assessment, Swedish University of Agricultural Sciences, Uppsala, 750 07, Sweden

[3] Retired, previously Swedish Meteorological and Hydrological Institute, Norrköping, Sweden.

*Correspondence to*: Jan Seibert (jan.seibert@geo.uzh.ch)

**Abstract.** Hydrological catchment models are important tools that are commonly used as the basis for water resource management planning. In the 1960s and 1970s the development of several relatively simple models to simulate catchment runoff started and a number of so-called conceptual (or bucket-type) models were suggested. In these models, the complex and heterogeneous hydrological processes in a catchment are represented by a limited number of storage elements and fluxes between these. While a major motivation for such relatively simple models in the early days were computational limitations, 15    today some of these models are still used frequently despite vastly increased computational opportunities. The HBV model, which was first applied about 50 years ago in Sweden, is a typical example of a conceptual catchment model and has gained large popularity over the past 50 years. During several model intercomparisons, the HBV model performed well despite (or because of) its relatively simple model structure. Here, the history of model development from thoughtful considerations of different model structures to modelling studies using hundreds of catchments and cloud computing facilities, is described. 20    Furthermore, the wide range of model applications is discussed. The aim is to provide an understanding of the background of model development and a basis for addressing the balance between model complexity and data availability, which will face hydrologists also in the coming decades.

## 1.    Introduction

The fundamental questions to hydrologists have remained similar over the past decades. How much water is available today 25    and will be in the future? How much discharge is there on average, at minimum and maximum flow? How extreme can it get? How does discharge vary between catchments? How do vegetation and climate influence discharge? What is the role of subsurface flow? How will river flow change over the days and months to come?

Despite these longstanding questions, many aspects of hydrological catchment modelling have changed over the past 50 years, as can be seen in the modified advice to hypothetical young PhD students:

Around 1970: "We need to simulate runoff – go and construct a hydrological catchment model."

Around 1995: "Use a physically-based model, not something as simplistic as this HBV model, which requires calibration to work well. Simple models will not be used for much longer. As soon as computers get more powerful, everyone will use more advanced models." (admittedly, already at that time, some researchers argued that simple models such as the HBV model are ideal for simulating catchment runoff while more complex models should mainly be used for applications where also other

variables are of interest (Beven, 1989; Refsgaard and Knudsen, 1996))

Around 2000: "Model predictions are necessarily uncertain. That's the way it is, so you better study how we can estimate that uncertainty."

Around 2020: "HBV for your PhD work – good choice, but why are you only using 100 catchments? How can you ever get any generally valid results with so few?"

In other words, doing a PhD in hydrological catchment modelling has remained challenging over the years, but the type of challenge has shifted from handling or sorting punched cards to making sense out of gigabytes of data. Imagine Sten, Jan and a student working on her (or his) PhD thesis discussing hydrological modelling (Fig. 1). Such a dialogue between PhD students from these different times talking about the HBV model (Fig. 2) might go something like this:

Student: "You used punched cards? How did that work? Was there an app for this?"

Sten: "In the very beginning, we even used paper tapes with the model code and data punched on it. If we were lucky, we could count on one or two model runs per day. By the 90s, computers became really powerful, and since then, we have run HBV on our desktop computers."

Jan: "Yes, I could even do Monte Carlo runs with a few million model runs, but I had to borrow all the department's computers over the Christmas break." (Seibert, 1999)

Student: "Only a few million model runs? This is what cloud computing does for me during the coffee break. By the way, you were calling HBV conceptual, bucket-type, physically-based, lumped and (semi-)distributed; now, what is it?"

Jan: "We have not always been consistent in model classification over the years. The term conceptual model is widely used in hydrology for these models, representing the basic concepts of water storages and fluxes. However, I now prefer using the terms bucket-type and semi-distributed. The first refers to the limited number of storage elements, the latter to the use of

elevation (and sometimes vegetation) zones as catchment area fractions."

Sten: "HBV is all of this. A simple model can be as physically-based as a more complex one. After all, in HBV we obey the water balance, which is a basic physical law. There have been endless discussions on which types of models are more physically-based (for an interesting exchange, see (Beven, 1990) and the subsequent discussion, later reflected on by, for instance, Beven (2001) and Refgaard et al. (2016)). But don´t forget that model complexity must match the input data available,

including all areal variabilities in a basin. For example, if we do not describe the albedo or atmospheric inversions reasonably well, we cannot expect a physically based snowmelt model to work either because the energy balance at the surface of the snow will be compromised. Honestly, I care more that a model provides useful simulations than intends to describe all processes in full detail. Performance counts! So, let's stick with conceptual and semi-distributed for now."

Student: "I read some of your papers and understand that you put some thought into model development, but what on earth were you thinking when deciding on the name HBV model?"

Sten: "Well, once we had the model, we needed a name for its first scientific publication. The name of the unit at SMHI where the model was developed (Hydrologska Byråns Vattenbalansavdelning) seemed like a good choice at the time. Had we only known how long we would stick to this name …"

Student: "And HBV light?"

Jan: "My version of HBV was designed for easier use in teaching. So, we started calling it HBV light and also used this name in an early publication. Each time I get an email such as 'I liked your light-model, can you send me the real one', I question the name."

Student: "And the performance measure R2, which is so often confused with $r^2$?"

Sten & Jan: "That was not us; blame Nash and Sutcliffe and their classic publication on hydrological modeling (Nash and Sutcliffe, 1970). We will come back to this later."

Student: "It seems a lot of researchers are using HBV these days, weren't there other conceptual models developed in the 70s-90s?"

Sten: "Yes, there were several. We will come back to them."

Student: "And today HBV is one of the most popular of these models; it seems I chose the best model for my PhD studies. What made HBV so successful?"

Jan: "In short, the 3 Ps"

Student: "What do you mean?"

Jan: "Parsimony - one thing I like especially about the HBV model is that the different model routines are all simple and use a limited number of parameters which need adjustments when tuning the model. Thus the frustration when trying to calibrate an overparameterized model can be avoided."

Sten: "Performance - the HBV model was able to do what was needed and performed quite well in several model intercomparisons" (Breuer et al., 2009; WMO, 1986).

Sten: "… and not to forget, persistence - it certainly helped that HBV was widely applied at SMHI and also became the main tool for operational hydrological forecasting in some other countries. In Sweden it also became the standard tool for the estimation of design floods (Bergström et al., 1992). Soon after its introduction, many researchers started and continued using the model for all kind of applications."

Student: "Interesting. Would you say there is anything one cannot do with HBV?"

Jan: "Models like HBV focus on the simulation of catchment runoff. If you are interested in the spatial variation of internal water fluxes and states, you might have to use a fully distributed model after all. Furthermore, quantifying impacts of climate change and land-use/-cover changes on hydrology as well as simulations for ungauged catchments are difficult …."

Sten: "… but we do it anyway. More seriously, these types of applications were, for a long time, considered impossible with a conceptual type of model like HBV. Still, later in this text, we will describe how HBV can contribute here nevertheless."

Student: "It seems that HBV has become a widely used, powerful tool. How did this all happen? What were the ideas driving its development? And who were the people behind this development? What did the journey of HBV look like, and how will it continue?

Sten & Jan: "Good questions! We will try to address these in this paper, in which we describe the history of model development from thoughtful considerations of the formulation of different model structures to modelling studies using hundreds of catchments and cloud computing facilities. We show that the HBV model performed surprisingly well during several model intercomparisons, and we discuss the wide range of model applications."

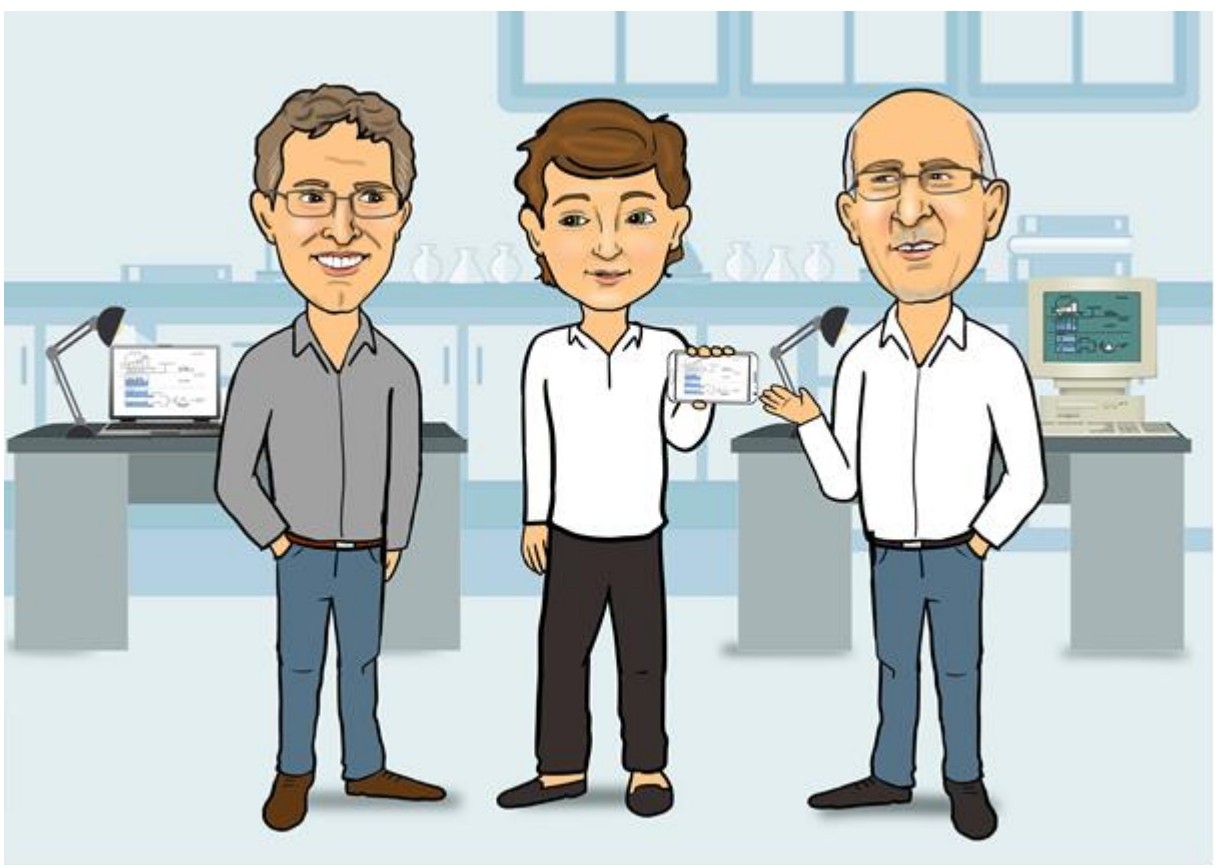

*Figure 1: Discussion of hydrological catchment modelling between Jan, a PhD student and Sten. The smartphone the PhD student is holding in her (or his) hand is far more powerful than the computers available when Sten started developing the HBV model in the early 1970s (drawing by Steph's Sketches).*

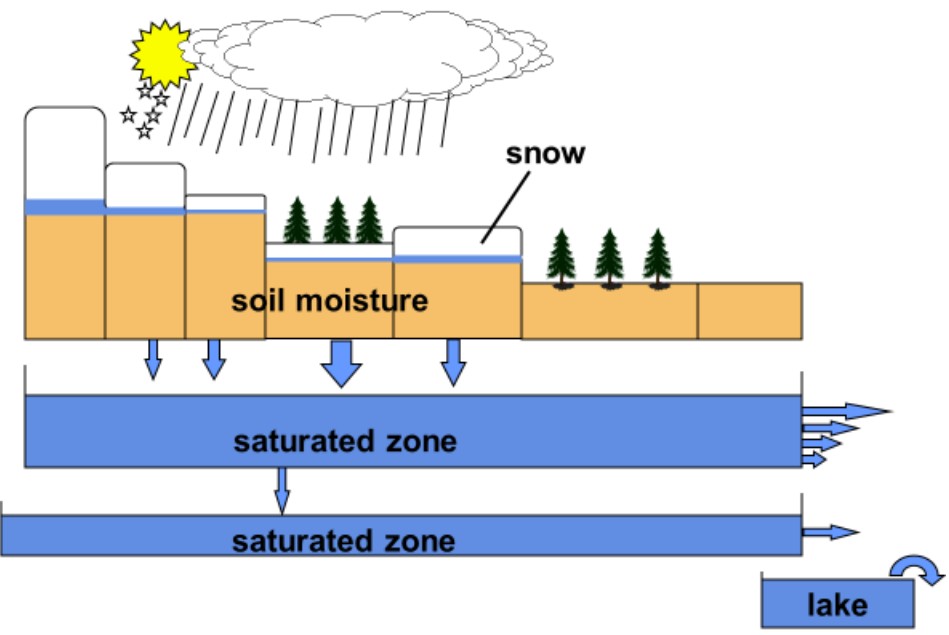

*Figure 2. Structure of the HBV model (drawn by Sten Bergström).*

## 2.    History of model development

### 2.1    Early thoughts on catchment modelling

The International Hydrological Decade (IHD), a research program initiated by UNESCO that spanned 1965 – 1974, became instrumental for advancing hydrology as a science, with new financial resources and intensified national and international cooperation. Computers had arrived, but in the beginning, they were only used by a specialised few. The first hydrological computer-based models, such as the SSARR model  (Brooks et al., 1975) and the Stanford Watershed Model (Crawford and Linsley, 1966) were developed already in the 1960s.

Internationally there was a lively debate on the principles in hydrological modelling. The initial enthusiasm about the calculation power of the new computers was met by arguments against the development of model systems that were too complex (Dawdy and O'Donnell, 1965). With too many details in a model, it was argued, the number of model parameters introduces so many degrees of freedom that it is difficult to handle and control the model.

In 1970 Eamonn Nash and co-authors published a set of papers that strongly advocated for best practices in model development (Mandeville et al., 1970; Nash and Sutcliffe, 1970; O'Connell et al., 1970). In the first paper, a strategy for model development was suggested under the heading "*Progressive modification*". Nash and Sutcliffe wrote in the end of their eminent 1970-paper

(p.290): "*If one accepts that it is desirable to have a simple rather than a complex model, and this is certainly true if it is hoped to obtain stable values of the optimised parameters, then it would seem that a systematic procedure would be as follows:*

*(1) Assume a simple model, but one which can be elaborated further.*

*(2) Optimise the parameters and study their stability.*

*(3) Measure the efficiency $R^2$.*

*(4) Modify the model- if possible by the introduction of a new part- repeat (2) and (3), measure $R^2$ and decide on acceptance or rejection of the modification.*

*(5) Choose the next modification. A comparative plotting of computed and observed discharge hydrographs may indicate what modification is desirable.*

*(6) Because all models cannot be arranged in increasing order of complexity it may be necessary to compare two or more*

*models of similar complexity. This may be done by comparing $R^2$."*

So, from the beginning, there were two fundamentally different approaches in hydrological modelling. On the one hand, start simple and add what is needed, and on the other hand, attempt to represent all relevant hydrological processes from the beginning, i.e., put in all we know at once. The development of the HBV model followed the first approach. It is important to note that while some modellers claimed (and partly still do) that more complex would provide a better representation of the

'true' processes in a catchment, there also have been contibutions from the early days on which argued that more complex models are only needed if other variables than catchment runoff is of interest.

The paper by Nash and Sutcliffe (1970) is one of the most cited in hydrological modelling, although this is mainly because of the introduction of the so-called model efficiency or explained variance. The authors called this measure $R^2$ but other names, such as NSE, are preferable to avoid confusion with the coefficient of determination. NSE has drawbacks and has been shown

to overemphasise errors in timing and magnitudes of flood peaks while less emphasis is placed on volume errors and low flows. Several alternative performance measures have been proposed over the years, but the NSE is still probably the most commonly used measure. Only recently has the KGE measure (Kling-Gupta efficiency, Gupta et al., 2009) gained in popularity. The advantage of this measure, and its modification NPE (Non-parametric efficiency, Pool et al., 2018), is that it allows different components of the total model error (correlation, bias and variability) to be distinguished.

Model calibration soon became an issue. As model calibration, where model parameters are successively adjusted until the best model performance is obtained, became a standard procedure, there was a risk that hydrological modelling degenerated into simple curve-fitting. The way to overcome this was to judge the model by its performance over a period that had not been used to calibrate its parameters. This split-sample testing became standard for most applications, including the international model intercomparisons carried out by the World Meteorological Organization in 1975 and 1986 (see below).

In the early days of model development the most trusted way of judging the performance of a hydrological model was to simply look at computed and observed hydrographs and to judge model performances by this so called visual inspection. Objective criteria of goodness-of-fit were used as support, but, for practical reasons, soon became the standard when the number of applications grew. The R2-value, as suggested by Nash and Sutcliffe in 1970, was the most popular of criteria,

supplemented with a measure of the volume error. The calibration process meant that model parameters were adjusted until a maximum value of R2 was found. This tuning was not a trivial problem as the limited capacity of the early computers did not allow a great number of simulations. Therefore various techniques to search for the optimum of the error function were used, such as the algorithm by Rosenbrock (1960). However, the number of model runs was still a limiting factor.

To ensure that the introduced model parameters really improved the model, the error function was sometimes mapped and its topography around optimum parameter values was analysed. A topography without a distinct peak at the optimum parameter values reveals parameters with low significance or severe interaction between them (Fig. 3).

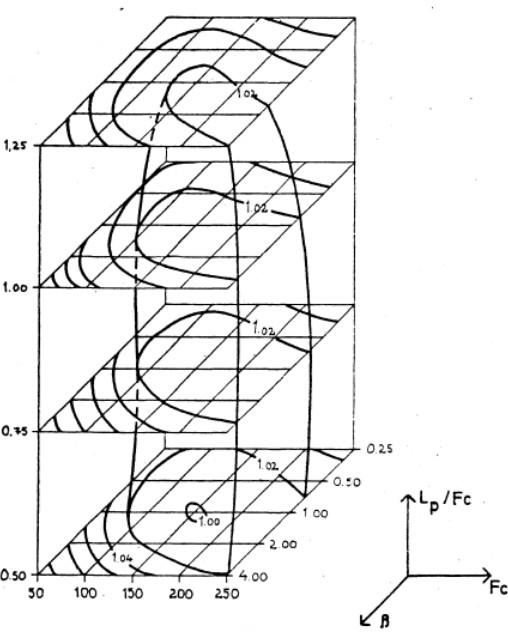

*Figure 3. Mapping of the error function around the optimum values of the three parameters in the soil moisture routine of the HBV model when applied to the Filefjell research basin in Norway. Hand-made drawing based on 100 model runs from Bergström (1976).*

In the early days of work on the HBV model we spent hours and hours simply looking at the computed and observed hydrographs, trying to understand what was really going on when trying to tune the model. Sometimes severe problems in the observations were also detected this way. During manual calibration typically 10-30 model runs were performed (note that in to 70s one model run took about half a day). Today automatic calibration of hydrological models is commonly used. Visual inspection, however, should not be ruled out in the development phase. It gives the hydrologist an insight into the dynamics and sensitivity of the model to disturbances in its parameters and indicates important interactions and helps in identifying feedback mechanisms between model components. It is time well spent.

## 2.2    Catchment model developments

Early models from the 1970s included the Canadian UBC model (Quick and Pipes, 1976), the Danish NAM model (Nielsen and Hansen, 1973), the Japanese TANK model (Sugawara, 1979), the Swiss-American SRM model (Rango and Martinec, 1979), the US National Weather Service River Forecast System (NWSRFS, Anderson, 1973), which is based on the Sacramento Catchment Model (Burnash, 1995), the GR4J model (Michel, 1983; Perrin et al., 2003) and the Swedish HBV model (Bergström, 1976). Later models included the British TOPMODEL (Beven and Kirkby, 1979), the Chinese Xinanjiang model (Zhao, 1992), the Danish MIKE-SHE (Refsgaard et al., 2010), the Italian ARNO model (Todini, 1996) and the American VIC model (Liang et al., 1994). An overview of some of the best-known hydrological models at the end of the 20th century can be found in the work by (Singh, 1995). Peel and McMahon (2020) provide a recent review on the development of runoff models.

While there has been an exchange of ideas with some model developers, there were also independent developments in parallel. The philosophy behind the development of the French model GR4J by Claude Michel (Michel, 1983; Perrin et al., 2001) has large similarities to the HBV philosophy, but despite developing the models around the same time, there was no exchange between the two model developers.

Early in the hydrological modelling debate, the term 'conceptual model' was contrasted with 'physically-based models', and 'lumped models' were contrasted with 'distributed models'. These are rather illogical classifications that have persisted over the years. It was understood that a conceptual model in this sense lies somewhere between a black-box approach and a fully physical representation of the hydrological system. Still, as almost all models are based on the water balance in some way, all models could be said to be 'physically based'.  Using such a broad interpretation of the term 'physically-based' implies that the classification becomes meaningless. There is an obvious difference between bucket-type models such as the HBV model and more sophisticated spatially distributed models such as the SHE model or the modeling system presented by Kollet et al. (2018), but we argue that the latter type of model is not automatically more 'physically-based' in the sense of being a (more) correct representation of the real catchment processes. Beven (1989) already pointed out that using effective parameter values at the grid-scale in physically-based models implies that these models are necessarily conceptual approximations; even if the Richards equation applies, it does not average linearly. Model classification can also be seen as a more academic question, whereas operational model users are more interested in the performance and feasibility of the model.

In the beginning, the simpler models regarded the catchment as one homogeneous unit, so-called lumped models, but today conceptual models are usually applied with some form of spatial variation. In mountainous catchments, for instance, the use of elevation zones is common and these models are also used with vegetation zones or subcatchments.  In other words, spatial variations are considered when describing fluxes and storages. While these cannot be mapped to exact locations in the catchment, the model represents spatial variability by distribution functions, and thus the term semi-distributed is used.

There are also gridded implementations of the HBV model. It started as a way of producing water balance maps and made it possible to better communicate and co-operate with climate modellers, who were used to gridded formats. The first gridded HBV model was used to produce hydrological maps for the National Atlas of Sweden (Raab and Vedin, 1995). Since then, simulations with gridded model versions have become more common. However, the grid cells are typically rather large (1 km$^2$ or more) to ensure that lateral groundwater flow between the cells is negligible (i.e., small in comparison to streamflow), and the groundwater storages in adjacent cells can thus be treated as independent of each other. It should also be noted that such a distributed version of the HBV model does not necessarily lead to better runoff simulations (Das et al., 2008). On the other hand, distributed versions of the HBV model have been used to calibrate parameter values simultaneously against data from many different catchments. This approach has been used, for instance, for generating runoff maps for Norway (Beldring et al., 2003) and Georgia (Beldring et al., 2017).

Using field observations of internal model variables such as snow pack, soil moisture deficit or groundwater dynamics to evaluate the performance of a catchment model is challenging. There is the commensurability issue (Beven, 2018) when comparing observed point-values to internal areal model variables. Direct comparisons are also tricky when using aggregated observation data or spatial observations (e.g., remote sensing of snow cover). Nevertheless, several attempts of comparisons of the internal variables of the HBV model against field observations have been carried out (Andersson, 1988; Bergström and Lindström, 2015; Bergström and Sandberg, 1983; Brandt and Bergström, 1994; Seibert, 2000). These studies suggest, at least qualitatively, that the dynamics of the internal variables of the model agree reasonably well with observations and thus help enhance the confidence in the model.

## 2.3    The story of HBV in Sweden

In Sweden, intense large-scale hydropower development had been going on since the beginning of the 20$^{th}$ century and was about to be completed in the 1970s. The hydropower system had become the backbone of the electricity supply system, but further development met strong environmental concerns and opposition. It was time to focus on the operation of the system rather than further exploitation of the rivers. The industry was now looking for a reliable and practical operational forecasting tool for inflow to reservoirs and power plants.

The Swedish Meteorological and Hydrological Institute (SMHI) had a long tradition of working with hydropower development. It also had the advantage of hosting meteorology and hydrology under the same roof, where the meteorological service could provide computer resources and meteorological data for the hydrologists. SMHIs small hydrological research unit, HBV (Hydrologiska Byråns Vattenbalansavdelning), was also a very active national part in the scientific co-operation within the International Hydrological Decade, IHD. So, the scene was set for computerised hydrological modelling at SMHI.

Eamonn Nash spent some time at SMHI in Stockholm in 1972, and his ideas came to have a strong impact on the hydrological modelling work at the institute. With operational use in mind, it was important that the complexity of the model could be justified by its performance and that data demands could be met even in remote areas with limited data coverage. In particular, the number of model parameters, which are subject to calibration, should be kept as low as possible to avoid parameter

interactions and subroutines representing insignificant process details. Today we call that overparameterisation. It was a

surprise how soon the point was reached where increased complexity did not improve model performance.

The first successful run by an early version of a hydrological model at SMHI was carried out in the 12.7 km² Lilla Tivsjön research basin in the spring of 1972 (Fig. 4). The results were surprisingly good with this model embryo with only one runoff component. Later applications showed that in general three components are needed for good agreement between modelled and observed runoff. Still the model performed surprisingly well in spite of a very simple structure and only few parameters to

calibrate. The model was then named the HBV model (from the abbreviation of the research unit where it was developed), and the first report (in Swedish) was published the same year (Bergström, 1972). Later that year, modelling results were presented to Nordic colleagues at the Nordic Hydrological Conference in Sandefjord, Norway. In 1973 the first publication in an international scientific journal appeared (Bergström and Forsman, 1973). It is interesting to note that simultaneously, and in the very same edition of Nordic Hydrology, the Danish NAM model was published by Nielsen and Hansen (1973). It is similar

to the HBV model in its simplistic setup, but subroutines are formulated differently.

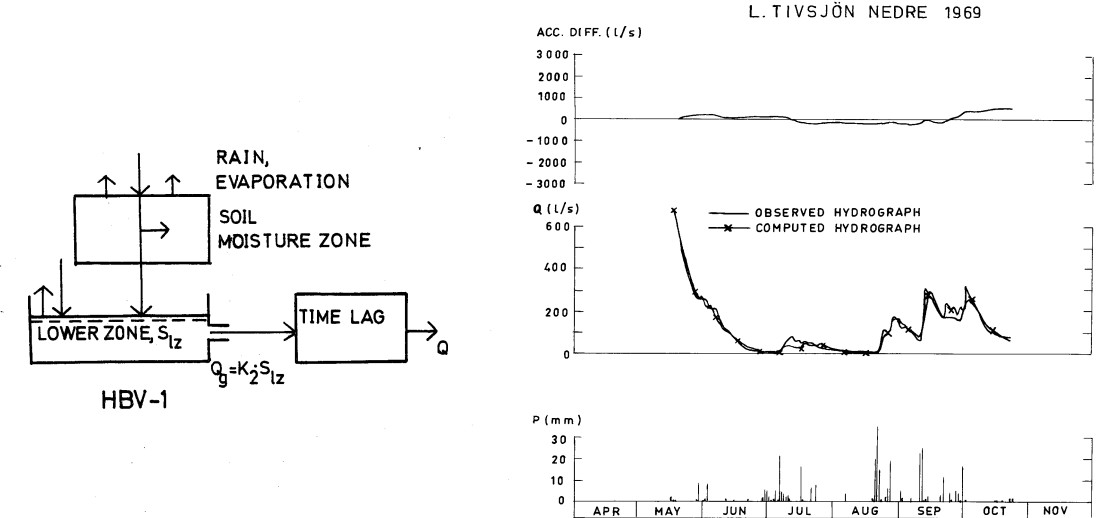

*Figure 4. A sketch of the very first version of the HBV model and the very first successful simulation of runoff based on data from the Lilla Tivsjön basin (from Bergström, 1976).*

In the pioneering days, when computerized hydrological modelling had just begun, working conditions differed a lot from today. Even if you had access to a computer, you could not count on more than a few test runs per day. At SMHI, for example, you programmed yourself, and you also had to punch the model code, input and verification data yourself on paper tapes or cards. Geographical Information Systems, GIS, were not heard of, so extracting the hypsometric curve and similar mapping work had to be done manually from a paper copy of a topographic map. When you were ready for a test run, you simply walked

into the noisy computer center and kindly asked the staff to run your work in between the meteorological forecasts. Visual

comparison of the graphs of the computed and the observed hydrograph was the most important way of judging the model performance. The R2 value was used as a support criterion of performance only.

The first tests with the HBV model were made for the snow-free period only. A first subroutine for snow accumulation and melt was added to the model in 1975 (Bergström, 1975). With the introduction of an area-elevation zoning of the snow routine

for mountainous terrain in 1976, the model was ready for operational tests (Bergström, 1976; Bergström and Jönsson, 1976). In 1976 the first operational forecasts were carried out for the hydropower industry. Both short-range (a few days) and long-range forecasts (several months) were carried out after the model had been updated to good starting conditions. The short-range forecasts were based on meteorological forecasts while the long-range forecasts used a set of historical climate records for the forecasted period to simulate a range of possible future river flows (Bergström, 1978).

Parallel to the work for the power industry, interest in flood protection grew, in particular after severe flooding in the spring of 1977.  Modelling started for some flood prone rivers in central and southern Sweden in the late 1970s.

The HBV model proved to perform well based on how it managed to simulate river runoff in an increasing number of basins, but in the beginning, its merits were more appreciated by engineers and power producers than academics. Maybe it was considered too much of an engineering tool rather than a scientific achievement. The necessary calibration process was

sometimes looked upon as mere curve fitting of a black-box model. It was evident that the hydropower industry and the national hydrological service, SMHI, had a more practical view on hydrological modelling, than scientists from universities. It took some time before the physical considerations and interpretation in the HBV model were appreciated by academics in Sweden. Nevertheless, the work on the HBV model resulted in a PhD thesis, which was defended in 1976 (Bergström, 1976), and is still the standard reference to the HBV model. Since then, various aspects of the HBV model have been addressed in numerous

PhD theses around the world. The continued popularity of the HBV model can also be seen from the steadily increasing number of publications (Fig. 5).

The hydropower industry realised early the potential of a model like HBV and strongly supported its further development. Hydropower operation thus became the main driving force in the early development of the model. The day-to-day dialogue with hydropower companies, where the model performance was investigated in great detail, was instrumental. The industrial

interest also helped to secure financial support for the model development and data collection.

It was soon realised that a production system with a practical and user-friendly interface is absolutely necessary for a hydrological model to become accepted as an operational tool. This system must include routines for model calibration, updating of the model as new data arrives and an interface for runoff simulation and forecasting. The development of the production system would prove to be a lot more costly than developing the model itself. The operational requirements

demanded the development of a user- friendly modeling system which, in turn, contributed to the widespread use of the HBV-model.

At SMHI the Integrated Hydrological Modelling System (IHMS) was developed and ready for use in the mid 1990s (e.g., Lindell et al., 1996). The model could now be run by almost anyone. At the same time, the commercial importance of HBV modelling grew. The source-code of the HBV model was made available to the academic community for research and

education upon request but with certain commercial restrictions. However, the advanced functionalities in the IHMS system, required from an operational point of view, were not needed by the academic research community and new model codes were written by several researchers.

The operational modelling system, IHMS, also included a routine for automatic model calibration. This was a necessity when the number of applications grew. The criterion of model performance, R2, suggested by Nash and Sutcliffe already in 1970,
was supplemented by a penalty for modelled volume errors in the automatic calibration routine in the IHMS-system (Lindström, 1997).

In the 1980s, several flood situations alerted the Swedish hydropower industry. The capacity of the spillways did not meet reasonable safety requirements and the risks for dam failures were unacceptably high. After five years of intense studies, new national guidelines for estimating design floods for the spillways were formulated and adopted in 1990. Emphasis was on peak
flows, but flood volumes were also of interest in the complex system of dams and reservoirs in the developed rivers. The HBV model played a key role in this process. The model was now developed for entire river systems, which meant that river regulation also was considered. By combining observed flood generating factors such as heavy rainfall, extreme snowmelt situations and wet antecedent conditions, the model was used to simulate floods which magnitudes far exceeded anything that had ever been recorded. These synthetic design floods were often more than twice as big as the highest observed floods, which
implied an extreme extrapolation far outside range during model calibration. This would never have been accepted in the model's infancy, but confidence in the performance had grown as the model was successfully applied in an increasing number of basins and under a wide range of climatological conditions.

In 1990, the new Swedish guidelines for spillway design floods were finally adopted and they have been applied ever since with only small modifications (Bergström, 1992; Flödeskommittén, 1990; Lindström and Harlin, 1992; Norstedt et al., 1992).
Today these guidelines are the standard procedure not only for the hydropower industry, but for all areal planning and developments in areas exposed to floods in the country. The safety of all major dams has been re-assessed, and many of them have been modified to meet the new standards. In 2015 the Swedish guidelines for design floods for dams were supplemented with a strategy for adaptation to climate change (Svenska Kraftnät et al., 2015).

The focus on extreme floods influenced the re-evaluation of the structure of the HBV model, which was carried out by SMHI
in the 1990s. The new model version, HBV 96, got a modified response function which performed better than the original one used thus far. This also meant that the number of free model parameters used for calibration of the model could be reduced by one, thus reducing the risk for overparameterisation, i.e., the inability to determine one single best parameter set (Lindström et al., 1997). It is interesting to note that the title of this paper is "Development and test of the distributed HBV-96 model." To the authors' knowledge, there have been no objections from the scientific community to using the word "distributed" in this
context.

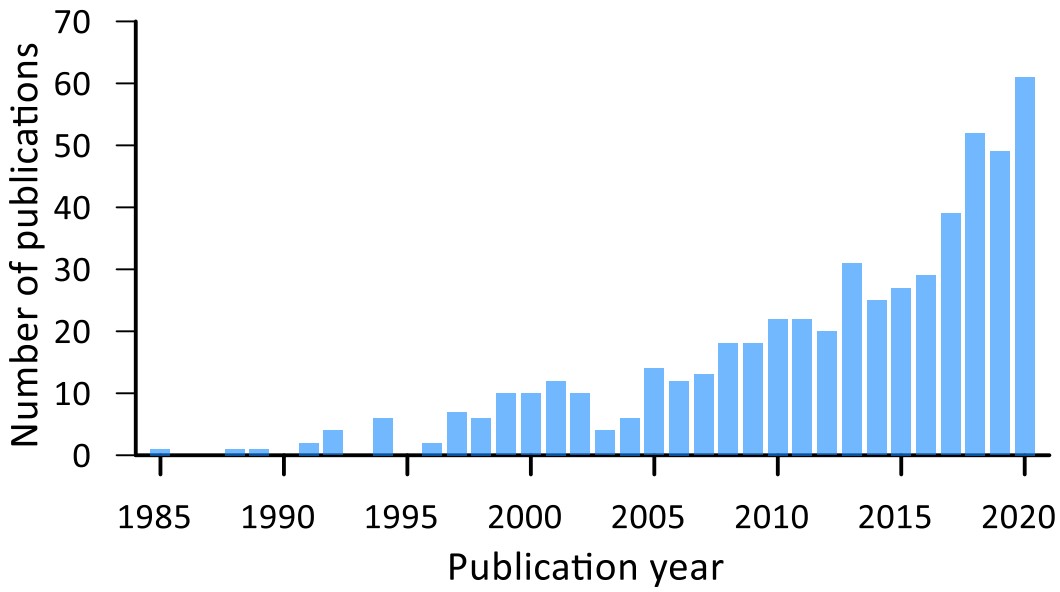

*Figure 5. Number of publications found in the 'Web of Science' using the search query "HBV AND (runoff OR hydrology)" (performed on 2.10.2021)*

### 3. The 3 Ps: Parsimony, performance, and persistence

The international success of the HBV model is in our opinion much related to the 3 P-s (parsimony, performance, persistence) as mentioned in the introduction. The parsimony made the model code easy to understand (and re-program), the HBV model often performed well especially in comparison studies, and it certainly helped that the model was intensely used by SMHI and the hydrological services of Norway and Finland, and became a standard tool for the Nordic hydropower industry (opposite to a single researcher or research group as was the case for other models). In this section we further elaborate on these points.

#### 3.1 Parsimony in model development

A key characteristic of the HBV model is its parsimony. In initial tests, parameters that were found to have little importance were eliminated. Compared to most other models, the HBV model thus has a rather low number of model parameters. The HBV model started with a simple lumped approach. The drainage basin was assumed a homogeneous unit and has gradually developed into a geographically distributed model where subbasins are the prime hydrological unit, and lakes can be modelled explicitly

In general, three distinct physical components can be identified in the structure of the HBV model; snow accumulation and melt, soil moisture accounting and runoff response, including groundwater dynamics. The capability to simulate the latter was evaluated with satisfactory results already in the early days of the HBV model (Bergström and Sandberg, 1983). Since inflow

and outflow of lakes has a strong influence on river flow dynamics, major lakes can be modelled explicitly. For parameters and equations used in the HBV model we refer to previous descriptions such as the one in Seibert and Vis (2012).

### 3.1.1 Snow accumulation and melt

The simple traditional degree-day approach is the most commonly used approach for snowmelt modelling in the HBV model. A water retention capacity of snow is used to capture the delay between melt and the time when the water leaves the snowpack.

Topography is usually accounted for by area-elevation zones and temperature and precipitation lapse rates. Land characteristics such as forest and open land can be separated, and a statistically distributed approach can be adopted to account for redistribution of snow in open terrain. Glaciers are sometimes modelled separately.

Of great importance for the development of the snow routine of the HBV model was WMOs international intercomparison of models of snowmelt runoff (WMO, 1986) where the results from the HBV model were compared to other international models.

One conclusion from that project is specifically worth citing: "*On the basis of available information, it was not possible to rank tested models or classes of models in order of performance. The complexity of the structure of the models could not be related to the quality of the simulation results*." (WMO, 1986, page 35).

Since the publication of the results by the WMOs intercomparison project, numerous attempts have been made to improve the simple temperature-index technique by introducing more physically-based approaches, where the energy balance of the

snowpack is accounted for at least to some extent. The results have been mixed or disappointing. A recent comprehensive study by (Girons Lopez et al., 2020) of snow model complexity versus performance, carried out for 54 basins in Switzerland and the Czech Republic, largely confirmed these results. Several model modifications were tested, which were argued to increase the realism of the model. However, only a few modifications resulted in improved simulations, and these improvements were overall small.

### 3.1.2 Soil moisture accounting

The calculation of soil moisture deficit and its impact on runoff generation was the key to the immediate success of the HBV model. At first, it seemed obvious that any hydrological model should have a component for surface runoff, even if the soils were far from saturated. But this was not supported by observations in the research basins in Scandinavia. Instead, a simple curve was introduced to relate runoff generation to soil moisture deficit using the modelled moisture state and one free

parameter, the Beta-value. It meant that the soil acts as a buffer for runoff generation. When the catchment is dry most of the rainfall or snowmelt will be used to fill up the moisture deficit in the soil. When it is wet, most additional water will recharge the response function and eventually contribute to river runoff. This approach, first developed in 1972, has become the signature of all variants of the HBV model ever since. It has proved very versatile in applications to catchments over a great span of climatological and physiographical conditions in the world.

This does not exclude that surface runoff on unsaturated soils may occur locally due to very intense rainfalls. However, local surface runoff of this kind is normally difficult to detect in the integrated river flow at the outlet of a drainage basin of a large catchment.

The key modelled variable in the soil moisture accounting of the HBV model is the computed soil moisture storage, SM, which is allowed to vary between 0 and the parameter value FC. The relation SM/FC varies from zero to one and is an index of the

wetness of the soil in the basin, which determines how much of the water becomes recharge, and eventually runoff, and how much remains in the soil. The soil moisture storage is depleted by evapotranspiration, which has its potential value when the soil is wet and gradually declines as the soil dries out. For potential evapotransipiration usually long-term monthly average values ('the twelve values') are used, which are based on direct measurements or, for instance, the Penman-Monteith equation. Modifying these values based on temperature anomalies has led to improved runoff simulations (Lindström and Bergström,

390 1992).

The soil moisture accounting of the HBV model can be seen as an attempt to account for both the total soil moisture in the basin and its areal sub-basin or sub-grid variability. The soil moisture accounting approach probably also contributes to the often good performance of the HBV model as this implies that the model is robust with regard to the temporal distribution of precipitation and (small) timing errors. Similarly, model simulations might be relatively robust to random errors in precipitation

amounts.

While the simple structure of the model allowed recoding, it can also be noted that some technical details were implemented differently. Most importantly, the soil routine of the HBV model is highly non-linear. In the original code, the differential equation was thus solved by adding one millimeter at a time as described by Bergström and Forsman (1973) and Bergström (1976). This detail has not been implemented in the same way in all model versions.

### 3.1.3    Response function

The role of the response function of a hydrological model is basically to give the simulated hydrograph a shape in time that agrees with observations. Traditionally this has often been obtained using a set of reservoirs or buckets, where different outlets are used to represent quicker or slower runoff components.

When developing the HBV model it was found that good agreement between the shape of the computed and observed

hydrographs could generally be reached with two buckets and three runoff components represented by three holes and percolation from the upper to the lower bucket. It was supplemented with a transformation function to account for further damping of river runoff in the basin. This approach is still the most commonly used response function of the HBV model internationally. However, as mentioned above, when the HBV model was re-evaluated by SMHI in 1996, a new routine was introduced, where the two drainage holes of the upper reservoir are replaced by a continuous function. This approach proved

to perform better particularly for peak flow simulations. Another advantage was that the number of parameters in the response function could be reduced from five to four (Lindström et al., 1997). This new routine is standard at SMHI while the original five-parameter version is still frequently used in other model versions.

The response function was subsequently modified again by SMHI. The aim with this modification was to better cope with the variable dynamics of winter and summer peaks. The idea was to account for the variation in groundwater levels and thus the extension of recharge and discharge areas between wet and dry conditions in the basin. It was an attempt to let the hydrologically active part of the basin gradually become smaller when the basin is drying out. Numerically this effect was obtained by linking the drainage curve of the upper reservoir to the calculated soil moisture state as described by Bergström and Lindström (2015). Thus it was possible to model a more peaked flood response for summers than for winters, without introducing further model parameters. This is a modification which is not so well known, but has shown to have potential to improve the simulation. It is an option in the HBV model package from SMHI.

## 3.2 Comparative tests of model performances

In 1969 the World Meteorological Organization initiated the project "Intercomparison of conceptual models used in operational hydrological forecasting" with ten participating rainfall-runoff models (WMO, 1975). It was followed by the projects "Intercomparison of models of snowmelt runoff" (WMO, 1986) and "Real-time intercomparison of hydrological models" (WMO, 1988). This set of projects meant a lot for the practical use of hydrological models and set the standards for model intercomparisons. As an example, the validation of the performance of the individual models was based on an independent period of data not used for calibration of the models, and was carried out by an independent team. The conclusions in the final report were agreed upon during a final conference between the participating modellers.

Thanks to the WMO projects on intercomparison of hydrological models, modellers from around the world had the opportunity to discuss practice and experience in hydrological modelling and to compare results. The projects confirmed that more complex models do not necessarily outperform the more simple ones with respect to catchment runoff simulations. In snow modelling, the simple degree-day method, based basically on a near-linear relationship between air temperatures and melt, was hard to outperform by more sophisticated models, which accounted for the complete energy balance of the snowpack. It was simply impossible to describe the basin-wide areal distribution of the terms in the energy balance with sufficient accuracy for a model.

About 20 years later, another model intercomparison workshop took place in Giessen, Germany (Breuer et al., 2009). Several groups were invited to apply their respective models to the Dill catchment, a 693 km² catchment north of Frankfurt, to simulate land-use change effects. In the end, ten models were applied ranging from fully distributed models such as SHE with hundreds of parameters to simple models such as HBV. The initial step was a model intercomparison, and it turned out that the HBV model performed best (in terms of NSE) for both the calibration and validation periods (Breuer et al., 2009). This was an unexpected result as with more complex models and more parameters, one should, in theory, be able to achieve better model fits. It is important to note that for the more complex models only few of the parameters were optimized during calibration. Furthermore, in such model intercomparisons not only the model but also the experience of the respective modellers might influence the results. Discussions at the workshop also revealed that the fact that data were received only a few weeks before the workshop might have contributed to this result. With the HBV model, there was no problem setting up the model in time,

whereas groups with more complex models had run into several problems. Still, the results demonstrated that in practice, simple models could outperform more complex models.

One argument in favour of more complex models has always been the need for calibration for conceptual models. The argument is that for more complex, fully-distributed models, it is possible, at least in theory, to determine model parameter values based on field observations without any model calibration. The view of the possibility to use models without any calibration has changed over the years, and today, most modellers would probably accept the need for at least some calibration, or tuning, even for more complex models (Hrachowitz and Clark, 2017; Refsgaard, pers.com.). A full discussion on whether model parameters in more complex models are observable or not is beyond the scope of this paper. Still, it is important to note that conceptual models can also be used in an uncalibrated mode by using ensembles of randomly chosen parameter values. Recently the performance of the HBV model and the SHE model has been compared for more than 300 catchments in the UK (Seibert et al., 2018b). The parameter values for SHE had been derived based on detailed information of catchment characteristics, whereas for HBV an ensemble of random parameter sets was used. The performance of the two models with respect to simulated discharge was, on average, similar.

## 3.3 International development and use of the HBV model

There has been a great amount of persistency in the development and application of the HBV model over the years, both in Sweden and internationally. In addition to the development and use of the HBV model in Sweden, as described above, the model also quickly gained international interest. The first from abroad to adopt the HBV model were colleagues from Norway. The first results from Norway with the HBV model already appeared 1976. Comparative tests of hydrological models were also carried out in Norway (Saelthun, 1978). The exchange of ideas between Sweden and other Nordic countries intensified. In Norway, with its rugged mountainous terrain, Killingtveit and Aam (1978) contributed with a distributed approach to snow accumulation, which also influenced the development of the Swedish HBV model. The HBV model became a standard tool for the hydrologists of both the Norwegian and Finnish hydrological services (Killingtveit and Saelthun, 1995; Saelthun, 1996; Vehvilainen, 1986). During the 1980s, the international expansion of the HBV concept accelerated (Bergström, 1992). The simple structure and ease of understanding and handling of the HBV model were appreciated and good results were reported from an increasing number of countries and modelling groups. The global spread was enhanced by developing an increasing number of new variants of the HBV model developed by the international scientific community.

New versions of the HBV model appeared in many countries, some with modified descriptions of the physical processes but the basic structure remained the same. Braun and Renner (1992) developed a Swiss variant named HBV-ETH, which was frequently used in alpine areas (Hagg et al., 2007). A German version, named HBV-D, was presented by Krysanova et al. (1999) and later Stahl et al. (2008) presented the Canadian HBV-CE version (Jost et al., 2012), which today is commonly used in particular in mountainous basins with glaciers (Northern Climate ExChange, 2014). Another version of HBV has been developed in Austria at the TU Vienna (Merz and Blöschl, 2004).

Another model version is HBV light, which was launched in the late 1990s by one of the authors (Seibert and Vis, 2012). The motivation for developing this version was to provide a user-friendly model version, especially for use in teaching. The user-friendliness was achieved by simplifying the model to its core functionality (e.g., leaving out functionalities related to hydropower operation) and providing a graphical user interface (GUI) (remember that GUIs were rather new at that time). In the meantime, HBV light has been further developed and includes features like automatic calibration, optional model variants, subcatchments and a dynamic glacier routine (Seibert et al., 2018a). Furthermore, it is possible to use different time-steps and to trace water particles through the model (Weiler et al., 2018).

The HBV concept has also been included in distributed model frameworks. A good example are model developments in Norway where a gridded version of the HBV model was combined with other process descriptions such as evapotranspiration modelling (Huang et al., 2019), streamflow routing (Li et al., 2014) and glacier ice melting and retreat (Li et al., 2015).

Today the number of countries where the HBV model has been applied, to the authors' knowledge, is on the order of 100. The applications often started with hydrological forecasting in mind, but nowadays, studies of climate change impacts on water resources and simulations in ungauged catchments are increasingly common.

## 4.  Challenging applications

The development of the HBV model started in rather small research basins and then moved to larger sizes, which were of more interest to the power industry. With the growing interest in climate change and its possible impact on water resources came attempts to model continental scales, like the drainage basin of the Baltic sea. At first, this seemed to be a tremendous task, but the flexible structure of the HBV model allowed such applications. It seems that many of the small-scale processes average out at larger scales and, thus, the simulated runoff often looks even better for very large basins than for very small ones.

The modeling of the drainage basin of the Baltic Sea was a truly distributed continental-scale model application covering some 1.6 million km$^2$ (excluding the area of the Baltic Sea itself) and with territories from 14 nations. The meteorological database consisted of daily observations of air temperatures and precipitation from some 800 synoptic stations which were interpolated to a 1x1 degree grid. The hydrological data were monthly values of river runoff from major rivers in the area, covering 86% of the catchment. Runoff from the remaining 14 % was found by interpolation using neighbouring stations. The HBV model was applied to 26 subbasins. Still, model calibration focused on five main subregional drainage basins (Bergström and Graham, 1998; Graham, 2004). This model application delivered input to the international Baltex research program.

Another important change in HBV applications is related to the increased computational power and availability of data. With increasing computational power, it has become possible, and expected, to quantify uncertainties. This can be done using Monte Carlo approaches, repeated model calibration trials and ensembles of suitable parameter sets. With the increasing availability of hydrological data sets large-sample hydrological modelling has become possible. There are, for instance, data sets with hundreds of catchments for the US, UK, Chile, Brazil and Australia, which include all data necessary to run the HBV model (e.g., Addor et al., 2017; Alvarez-Garreton et al., 2018; Chagas et al., 2020).

While the use of the HBV model is straight-forward for many hydrological questions, answering some of the questions posed in the introduction remain challenging. These questions are mainly related to three types of model applications, land-cover change impacts, ungauged basins and climate change impacts, as discussed in the following.

### 4.1    Land-cover change impacts

Quantification of land-cover changes on hydrology is an important field of hydrological modelling. Here it is important to distinguish between change prediction and change detection. It is difficult to use conceptual models for change prediction because model parameter values are not directly linked to certain types of land-cover (Seibert and van Meerveld, 2016). For change detection, however, the HBV model has been proven to be useful (Brandt et al., 1988; Seibert et al., 2010; Seibert and McDonnell, 2010) by using the model as a control run, i.e., to simulate the discharge that would have been observed if there had been no change in the catchment. Another approach is the simultaneous calibration of the model to numerous catchments (Hundecha and Bárdossy, 2004). Such a regional calibration is possible for simple models such as the HBV model and allows a space for time approach to quantify land-use effects on catchment runoff using the HBV model.

### 4.2    Ungauged catchments

The most demanding type of application of a hydrological model is probably to use it without any calibration at all. This is referred to as prediction in ungauged basins. In the beginning, severe doubts were expressed when a conceptual hydrological model was used for this purpose, due to its need for calibration. Today, however, this is a standard type of application. The need for information about water resources is often so urgent that ungauged model results are accepted. The key to improving the reliability of the results is to generalise and use standard sets of model parameters from a vast number of applications. While there might be some relationships between catchment properties and individual parameters (Seibert, 1999), spatial proximity is often more suitable than the use of catchment properties for determining model parameters for ungauged catchments (Merz and Blöschl, 2005; Oudin et al., 2008). In general, transferring entire parameter sets from donor catchments to ungauged catchments results in better simulations than the use of regression equations (Oudin et al., 2008). The model performance will, of course, be lower than if the model is calibrated, but the results are often acceptable as they are better than no information at all. So even though the HBV model was primarily developed for runoff simulation after calibration, today it is  routinely used in ungauged basins as well (Bergström, 2006). Using ensembles of random parameter sets can result in surprisingly good model results. An indication of the value of using ensembles for HBV model simulations is described by Seibert and Beven (2009) who found that the ensemble mean performed generally better than the best single parameter set (see their Figure 2). This finding was recently confirmed in studies using large numbers of catchments (Pool and Seibert, 2021; Seibert et al., 2018b). Even better results can be obtained by ensembles of parameter sets that have been calibrated for other catchments (Seibert et al., 2018b). However, the question how to identify optimal donor catchments is still not fully solved (Pool et al., 2021).

## 4.3 Climate change impacts

The late 1980s saw the start of a new era in natural sciences. IPCCs scientific assessments were instrumental when climate change became the new focus for many large scale scientific programs. Climate change and its impact on water resources was identified as a major concern and also became a main focus in hydrological modelling. A number of extreme hydrological situations in various parts of the world emphasized this further. This meant a new role for the HBV model. A joint Nordic project on climate change impacts on runoff and hydropower in the Nordic countries was carried out (Saelthun et al., 1998). It was to be followed by similar projects where, eventually, the Baltic states also participated (Beldring et al., 2006; Nordic Council of Ministers, 2011). In Sweden, the Swedish Regional Climate Modeling Programme, SWECLIM (Rummukainen et al., 2004), and the establishment of its modelling centre, the Rossby Centre, was a breakthrough. New sets of regional climate scenarios were developed. These were made available for the hydrological scientists, who could also benefit from the large international network of climate researchers. The inter-disciplinary scientific dialogue between meteorologists, hydrologists and oceanographers intensified under several international research programmes. Assessments of the impact of climate change on water resources using global climate models, dynamical downscaling and hydrological modelling with the HBV model became practice (Bergström et al., 2001). In an early application, Beldring et al. (2002) used a gridded HBV model both for hydrological mapping in Norway and for simulations of the impact of climate change on the country's water resources (Beldring et al., 2006).

Today the HBV model is routinely used to model the impacts of climate change on water resources all over the world such as Himalaya (Akhtar et al., 2008),  the Ourthe (Driessen et al., 2010), the Rhine (Görgen et al., 2010), Finland (Veijalainen et al., 2010), the Nile (Booij et al., 2011), Sweden (Teutschbein and Seibert, 2012) or Canada (Stahl et al., 2008). One critical question asked when using a hydrological model for climate change studies was whether its parameter values, derived for one climate, could be used for the future climate.

An important question in this respect is the transferability of model applications between different climatic conditions. Differential split-sample tests have indicated that one has to be aware of additional uncertainties when parameter values that have been calibrated based on data from, for instance, cold and dry conditions to warm and wet conditions (Dakhlaoui et al., 2017). Another challenge is that climate change might affect catchments in several ways. Over longer periods, for instance, the treeline may change. This vegetation change will influence catchment functioning and should be reflected by changed model parameter values. However, in most climate impact studies, the hydrological model parameters are assumed to remain unchanged.

The use of dynamical downscaling of climate scenarios as input to a hydrological model for studies of climate change on water resources, means taking advantage of a chain of scientific efforts and the engagement of thousands of scientists. But it also means that the overview of the whole modelling process is lost. No one has the full detailed knowledge of all components in the chain of results from global climate models via regional climate models to hydrological models. One important reflection about inconsistency in the modelling process is that each one of the three modelling concepts (global climate model, regional

climate model, hydrological catchment model) has its own interpretation of the hydrological cycle. We leave the important task of harmonization of the modelling of the water cycle in these three categories of models to the coming generation of 575 modellers.

## 4.4 Model calibration and uncertainties

The three applications discussed above; land-cover change, ungauged catchments and climate change, cover important topics that are directly relevant in practice. However, HBV is also used for research on topics such as model calibration, uncertainty quantification and the value of data (e.g., Seibert, 1997; van Meerveld et al., 2017). Here, the HBV model provides an easy-580 to-use and computationally efficient model, which is a suitable representation of commonly used models. While many of these studies are more curiosity-driven and address basic issues, addressing these questions also provides the basis for good model application procedures in more practical applications.

A promising development are model applications using multi-criteria evaluation, i.e., the evaluation of model performances on more variables than just catchment runoff (Bergström et al., 2002). New sources of data (including spatial patterns derived 585 from remote sensing) might open new opportunities (Stisen et al., 2018). Still, there is a challenging balance between more available data and more model complexity to make these data directly usable for model evaluation. While this applies to any type of catchment model, fully distributed, complex models might provide more opportunities to compare field observations directly with model simulation. In contrast, in bucket-type (or conceptual) models, such observations need to be aggregated or otherwise pre-processed to become applicable (Seibert, 2000; Seibert and McDonnell, 2002).


## 5. Concluding dialogue

So, let us come back to our hypothetical discussion with the PhD Student.

Student: "Cool, this was an interesting lesson in the history of hydrological model development. What has been most surprising with HBV for you?"

Sten: "That it is still in use after all these years. That it has spread world-wide and that its performance is still competitive."

Jan: "That I am still using it after all these years. Remember that I started by showing how uncertain the HBV model was."

Student: "So, what happened?"

Jan: "A model can be uncertain and still be useful."

Sten: "Yes, a model does not have to be perfect, it needs to work under real-life circumstances like with limited data coverage.

And its performance must match requirements. I would also like to recommend any young modeller to spend some time in the field in the basin subject to modelling, just to get a feeling for Mother Nature's wonderful complexity. "

Student: "Is there any way we can improve the HBV model?"

Sten: "Sure, there were improvements over the past decades, and with new data becoming available, there might be new options."

Jan: "However, it is difficult to improve runoff simulations beyond the performance of relatively simple models such as HBV. Input data limitations are still one of the greatest obstacles."

Student: "These days, data for hundreds of catchments is made available. Would this not allow us to decide on the best model variant easily?"

Jan: "In theory, yes. The large sample hydrological modeling allows models to be evaluated in many more catchments, which

leads to more robust results. On the other hand, performance differences are often small and different model variants perform best in different types of catchments. Therefore, in practice, it is difficult to demonstrate that certain model variants are better than others."

Sten: "I am glad to see that a reasoning hydrological modeller is still difficult to beat."

Student: "What about data other than streamflow? Won't this allow us to develop better models?"

Jan: "Again, in theory, yes. One could argue that a model (and parameterization) that represents not only streamflow but also internal variables such as snow accumulation or groundwater storage realistically, is a better model and might provide more reliable simulations, especially when extrapolated beyond the conditions during calibration."

Sten: "But on the other hand, for most additional variables, model modifications are required to make these variables comparable with model simulations. This might result in increased parameter uncertainty and model complexity."

Jan: "This is an interesting balance and finding good ways to incorporate additional types of observations certainly requires hydrological creativity. This is especially true for novel types of observations such as remotely sensed data or citizen science observations."

Sten: "So in some ways not much has changed in the past 50 years, sound hydrological reasoning is still needed to decide on the most useful model structures and model testing."

Student: "Ok, but to be on the safe side we could represent all details of a catchment into our model. With all the increasing computational power, this should be possible. Wouldn't a fully realistic model not be great?"

Jan: "Not so sure, we are still limited by the available data to run our models. Even in places where there is a lot of data, there is often a mismatch in the scale of observations and simulations."

Sten: "… and take a look at these two models of an airplane, one fully detailed scale model and a simple paper airplane (Fig. 6).

What would you say, which model will fly?"

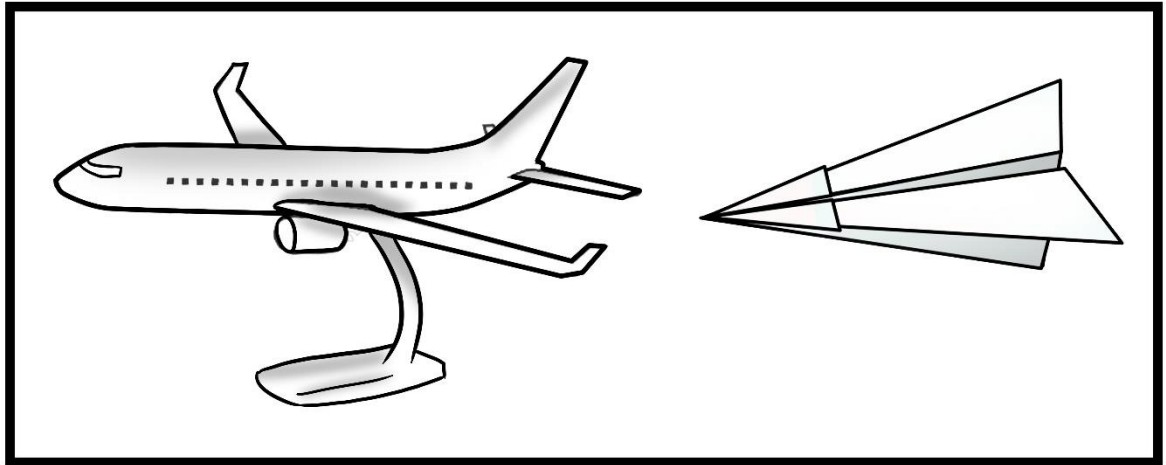

*Figure 6. Which modell will fly? (drawing by Steph's Sketches).*


**Acknowledgements**

We thank all the many 'HBV-colleagues' for stimulating studies and discussions over the years. We greatly appreciate the constructive comments by Keith Beven, Jens Christian Refsgaard, Axel Bronstert and András Bárdossy. Tracy Ewen, Sandra Pool and Marc Vis provided valuable comments on an earlier version of this contribution.

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
