# Peer review of "A retrospective on hydrological catchment modelling based on half a century with the HBV model"

_Hydrology and Earth System Sciences, 2021_

## Referee Comment (RC1)

**Review HESS – Seibert-Bergström, A retrospective on hydrological modelling based on half a century with the HBV model (HESS-2021-542)**

GENERAL COMMENTS

The manuscript describes the history of the HBV model development and applications. As documented through its widespread international use and the many scientific publications, it is unquestionable that HBV has made significant impacts on hydrological science and water resources practitioners. Hence, this is a success story that is definitely valuable to publish and learn from. And as such I believe that the topic and the story is relevant for the readers of HESS.

My main general comment relates to the writing style and the level of scientific discussion. The writing style is somewhat "easy going" with chapters 1 (Introduction) and 5 (Concluding dialogue) written as an informal dialogue between the two authors and a hypothetical PhD student. Furthermore, the manuscript is to a large extent "inward-looking" in the sense that it tells the HBV story without much critical self-reflection and without much scientific discussion of competing developments and discourses. While this probably makes the manuscript easier to read as a memoir/biography, it also reduces the scientific value.

Furthermore, as can be seen in my specific comments provided below, I find that some of the statements in the manuscript are conditioned on the relatively narrow scope (HBV story), which for instance makes some of the concluding dialogue appearing as general statements without proper reservation that the perspective deals with the HBV model (and similar rainfall-runoff models) and not hydrological modelling in general.

SPECIFIC COMMENTS

1. In my view HBV is a rainfall-runoff model and as such one of several types of hydrological models. To me hydrological modelling is not, as indicated in the first paragraph of the Introduction, confined to simulation of river discharge. Other types of hydrological models include plot/field scale models of the unsaturated zone, groundwater models, land surface models and coupled surface water/unsaturated zone/groundwater models. The authors talk about the HBV model as a hydrological model. I do not disagree to this. But it should be made clear that the statements in the manuscript on "hydrological modelling" are confined to "rainfall-runoff modelling". In line with this I suggest that the term hydrological model in the title be substituted with rainfall-runoff model.

2. The aim of the manuscript, most clearly stated in the abstract, "is to provide an understanding of the background of model development and a basis for addressing the balance between model complexity an data availability". I do not think that this is well elaborated in the discussion and in the concluding dialogue with respect to hydrological modelling in general. I could agree that this is

addressed in the narrow context of the HBV and similar model types, but the manuscript does not deal with this issue for other model types. This limitation must be stated explicitly in the manuscript.

3. Lines 30-32. The authors state that around 1995 the typical argument was "Use a physically-based model, not something as simplistic as this HBV model …". While I agree that such arguments existed at that time, many other scientists argued differently, namely that HBV type models were ideal for rainfall-runoff modelling, while physically-based models would mainly be useful for other applications – see e.g. Beven (1989), Refsgaard et al. (1992) and Refsgaard and Knudsen (1996). I am not arguing that this aspect should be mentioned here, but if the authors want to go into the scientific dispute between modelling schools that existed these days, it should somehow be considered.

4. Lines 86-88: This discussion is again confined to catchment scale, surface water hydrology. I can find many other examples on hydrological questions that cannot be addressed with HBV.

5. Chapter 2. Sections 2.1 (the early days of hydrological modelling) and 2.3 (the story of HBV in Sweden): Interesting and well written sections. I have some difficulties with section 2.2 as reflected in the next two comments.

6. Lines 171-179: The listing of "early models from the 1970s" does not reflect the heading "early types" – the models are not linked to different types.

7. Lines 184-205. The manuscript here introduces and discusses the typically used classification terms conceptual versus physically-based and lumped versus distributed. Without introducing rigorous definitions, the authors end up concluding that there is hardly any difference and that the HBV can be considered distributed and physically-based. Depending on the definitions one chooses this may be correct, but if you choose so broad definitions the classes become meaningless. I acknowledge that HBV can be used in a semi-distributed (HRU type) manner and that there is a physical basis behind most of the equations in HBV. But I shall also claim that there are fundamental differences between a model based on HBV and a very sophisticated spatially distributed, physically-based modelling system, for instance like the one described in Kollet et al. (2018).

8. Subsection 3.1.2 and lines 352-353: I completely agree and would consider the soil moisture accounting and the split between fast/surface runoff and infiltration to be the heart of a model of HBV type.
   - In fact, there are many similarities between all good performing models of this type with respect to how they handle this – the equations may appear different, but substantially the models act the same way.
   - Another reason for the success of the soil moisture accounting is probably that it is robust towards the temporal discretization of precipitation. The first versions of HBV operated with daily time steps – and you do not generate Hortonian overland flow from 24-hours rain but from high intensity rain over shorter time, e.g. minutes or a couple of hours. The

HBV (and the other similar models) have proven to perform well both when using daily and hourly rainfall – i.e. it is robust for temporal lumping of rainfall.

I am not suggesting text changes here, but if the authors want to discuss one of the reasons for the success of HBV and similar models, this aspect should probably be emphasized.

9.  Lines 393 -403, Intercomparison project (Breuer et al., 2009). I have a couple of reservations about the way this study is referred to:
    - "SHE with hundreds of parameters to simple models such as HBV". It is correct that distributed models have many parameters, but parameterization is always done in such a manner that most parameters are not modified during calibration. In this study, 7 parameters were calibrated for SHE, while 10 parameters were calibrated for the HBV. So, I think the text is potentially a bit misleading here.
    - Although Breuer et al. (2009) concluded that "there was no superior model if several measures of model performance are considered", I agree that it is correct that HBV showed the best performance with respect to NSE. Based on experiences from other intercomparison studies, e.g. WMO (1988) and Karlsson et al. (2016) I would argue that the differences in performance might be explained by the skill of the hydrological modeller rather than the quality of the model code.
    - Lines 401-402: I agree that the results from Breuer et al. (2009) suggest that simple model can perform at least as well as complex models with respect to runoff simulations. But the qualifier "with respect to runoff simulations" is important, because the results from Breuer et al. (2009) do not go beyond this.

10.  Lines 403-405: It is correct that such statement has been made, especially during the 1980s, but my impression is that the overwhelming part of the hydrological modelling community would argue that all hydrological models, including complex models, need some kind og calibration (Hrachowitz and Clark, 2017). So, I do not find that statement representative.

11.  Section 3.3: The international use of HBV has been an indisputable success story. I would argue that several other models of the same type had similar qualities when they were originally developed, but were not maintained and kept alive for several decades as the HBV. It would be interesting to hear the authors' evaluation of why HBV succeeded in this respect.

12.  Lines 545 – 554: I agree to this, when the statements are conditioned on HBV. However, this does not apply for more complex model types, where there are many examples of models calibrated against several target variables (e.g. field data on discharge, soil moisture, groundwater heads, land surface temperature, evapotranspiration) result in models with a more realistic representation of the hydrological processes, e.g. Stisen et al. (2018). I would also argue that the emergence of huge amounts of high-resolution good quality data in these years are likely to change the conclusions to this question.

REFERENCES

Beven, K. (1989). Changing ideas in hydrology – the case of physically based models. Journal of Hydrology, 105, 157-172.

Hrachowitz, M., & Clark, M.P. (2017). HESS Opinions: The complementary merits of competing modelling philosophies in hydrology. Hydrology and Earth System Sciences, 21, 3953-3873.

Karlsson IB, Sonnenborg TO, Refsgaard JC, Trolle D, Børgesen CD, Olesen JE, Jeppesen E, Jensen KH (2016) Combined effects of climate models, hydrological model structures and land use scenarios on hydrological impacts of climate change. Journal of Hydrology, 535, 301-317.

Kollet, S.J., Gasper, F., Brdar, S., Goergen, K., Hendricks-Franssen, H.J., Keune, J., Kurtz, W., Küll, V., Pappenberger, F., Poll, S., Trömel, S., Shrestha, P., Simmer, C., & Sulis, M. (2018). Introduction of an experimental terrestrial forecasting/monitoring system at regional to continental scales bases on the Terrestrial System Modeling Platform (v1.1.0). Water, 10, 1697.

Refsgaard JC, Seth SM, Bathurst JC, Erlich M, Storm B, Jørgensen, GH, Chandra S (1992) Application of the SHE to Catchments in India - Part 1: General Results. Journal of Hydrology, 140, 1-23.

Refsgaard, J.C., & Knudsen, J. (1996). Operational validation and intercomparison of different types of hydrological models. Water Resources Research, 32 (7), 2189-2202.

Stisen, S., Koch, J., Sonnenborg, T.O., Refsgaard, J.C., Bircher, S., Ringgaard, R., & Jensen, K.H. (2018) Moving beyond runoff calibration - Multi-variable optimization of a surface-subsurface-atmosphere model. Hydrological Processes, 32(17), 2654-2668.

---

## Referee Comment (RC2)

Review of the manuscript

A retrospective on hydrological modelling based on half a century with the HBV model

(Authors: By Jan Seibert and Sten Bergström, submitted to HESS)

Dear colleagues,

This manuscript tells us a story of the history of the HBV-model, in connection with some reflections on the development of hydrological modelling over the past decades. The manuscript is well written, nice to read and embedded into a fictional interview of a "young" (i.e. less experienced) PhD-student. This is a funny and unusual style-method to make a story more entertaining, which otherwise might be a bit less exciting for outsiders.

I find the manuscript interesting and it contains and summarizes various information regarding the HBV-model. Of course, this information are not "new" in the sense that they have not been presented before. However, as this article is planned as a contribution to a special Issue on 'History of hydrology', it fits very well into this overall theme.

I have some more detailed and minor comments (see below) and three more general remarks and suggestions. If those are followed / incorporated into the article, the final article could lead towards a broader discussion and thus have a more relevant impact.

General remarks and suggestions:

1) **"Physically based" vs. "conceptual" ?**
I think that the 'modelling philosophy' presented here is a bit narrowed to the personal view of the HBF developers, and to the state of discussion in the 1990ths. For example, when I was a young PhD student (I am in something between the ages of the two authors), I always followed the discussion about which models "are allowed" to be called "physically based" and which are "only" considered "conceptual" with some incomprehension but also amusement. My feeling was – and still is – that this is an artificial or at least not a meaningful distinction, which is not really worth to follow. Why? First, in principle, all the mentioned models follow the basic physical principles of mass and energy conservation. The authors also claim this in their manuscript and that is why they demand that HBV also be classified as a physically based. But second, there can always be a higher degree of physical detailing. Thus, even the "most physically based" model will have shortcomings regarding some peculiarities of the hydrological cycle or some hydrological processes.

Therefore, I think that the more relevant questions are: which practical tasks can a hydrological model solve and which hydrological processes can be distinguished and quantified? Therefore, I would recommend to better use the term "process oriented model" rather than "physically based". It just makes more sense. Following this line, one can elaborate on which hydrological processes are represented in the model by which approaches. In this regard, HBV is a model with a focus on snowmelt processes and runoff generation as a consequence of catchment wetness (a subsurface hydrological process). It is not made for, e.g., hydrological conditions, where soil-moisture-evapotranspiration

interactions are key, or – another example – where Hortonian overland flow is the main runoff generation process.

**So my recommendation in this regard: on should more clearly state, for which questions/tasks and hydrological conditions the model is well suited, and for which less.**

2) **Number of parameters and desire for optimization**

Another issue, which is related to this issue: One should not compare the number of parameters of lumped and distributed models. This is comparing apples with oranges. The authors are proud that HBV has only 5 (or even 4) parameters and compare that with the well-known SHE-model (lines 395/396) stating that "such as SHE with hundreds of parameters". These "hundreds of parameters" are a consequence of its areal distribution. If HBV is run in a distributed mode (what is possible and stated by the authors), it can also have hundreds of parameters. Of course, when one looks on one "SHE-grid" only (i.e. SHE in a nun-distributed mode), that SHE has much more parameters (may be around 20 – 30?). The reason is that it entails more processes and a higher degree of detailing.

A few parameters are advantageous for optimization procedures, but not necessarily sufficient to find a clear optimum (as shown many times by Keith Beven and followers). So, if an optimized model (usually optimized for discharge at the catchment outlet only) is desired, a low-parameter model is the best choice. However, if detailed spatial patterns and /or various processes are aimed for, one needs a higher degree of distribution and detailing of processes. This will inevitably yield (much) more parameters, lesser Optimization capability and the "danger" of equifinality. Still, one should keep in mind, that also in the hydrological nature certain states (or hydrography) might be reached by different potential conditions and system states. Thus, why should our models behave in a more convergent than nature does?

Finally, I find it absolutely essential to discuss the issue of which process should be described well by our model. The authors say that HBV performs well, several times. I agree, if one looks on the runoff dynamics at the catchment outlet. But I would not recommend this model for some other purposes, e.g. for (areal patterns of) root water uptake and evapotranspiration, or let's say surface runoff and subsequent erosion.

**Thus, I think model purpose and why and why not distributed models might be appropriate needs to be discussed. In this regard, an important purpose is also the (easy and/or quick) applicability. One of the major strengths of HBV.**

3) **Need for a classification of hydrological models**

Based on the previous thoughts, instead of comparing between different hydrological models in a competitive manner, I would ask for a kind of classification / categorization along the question, which tasks should be solved and on which processes the focus of the model lies. In this regard, I would claim there is a "family" of models where the runoff response of a catchment is simulated through a non-linear function of the catchments wetness. HBV is probably the most prominent member of this family. Others, most already mentioned by the authors, belonging to this family and about the same age are NAM (Nielsen & Hansen, 1973), Xinanjiang model (which, was 1st presented already in 1977 in China (Zhao, 1977) and than internationally in 1980 (Zhao et al, 1980)), VIC-model (Liang et al. 1994), GR4J (Michel, 1983). Even the famous TOMPODEL has ties to this family, because the overall runoff response of a catchment is also related to a non-linear function,

which is derived, however, from the catchments DTM. Other similar models have founded the sons and daughters of this first generation, such as the Arno-model (Todini, 1996) or in Germany LARSIM (Ludwig & Bremicker, 2006). And many others.

In this regard, the first generation of this family, with HBV as its best-known member, has been very successful. Still active and in use, and so many children and grand children.

But one must also be aware other families, which are able to contribute significantly to some of today's hydrological tasks, e.g. the ones which focus more on (Hortonian) surface runoff generation, such as KINEROS (Smith et al, 1995), QSOIL (Faeh, et al., 1997), HLIIFLOW (Bronstert & Plate, 1997) or its extension CatFLow (Zehe et al., 2001)

And there are more such families, as the ones concentrating on groundwater dynamics, urban hydrology and the ones which try to combine as much as possible, such as the previously mentioned SHE-model, or focus on agricultural landscape such as the most applied SWAT (Arnold et al, 1994) or its younger sister, the eco-hydrological SWIM (Krysanova et al, 1998).

**Now, let that be enough of categorization. It would certainly be profitable if the authors could include some of these aspects in their manuscript.**

**Some detailed comments:**

- Line 115: "advocation for realism in model development"? What is 'realism' in this regard? And Is this still valid for today?
- Line 130: Instead "represent all hydrological processes", better term it "represent all relevant hydrological processes",
- Line 285: what do you mean with "re-organising observed flood generating factors'?
- Line 300: "risk for overparameterisation"? You may discuss it in the relation with complexity of model and nature? Also nature may have several means to come to one state? What can one do in this regards?
- Line 315: "groundwater dynamics"? You may elaborate a bit how far the catchments wetness in the HBV model can be related to observed gw-dynamics.
- Lines 465-480: about the application for ungauged catchments: You mention the option of ensemble applications (many sets of parameters). You may elaborate a bit more on this. For my understanding this is a very valuable approach.
- Figure 6: The value of a model depends on its purpose!! If the purpose is to look like the reality, the left figure is much "better".

**Mentioned literature:**

Arnold, J.G., Williams, J.R., Srinivasan, R., King, K.W., Griggs, R.H.(1994): SWAT, Soil and Water Assessment Tool, USDA, Agriculture Research Service, Grassland, Soil & Water Research Laboratory, 808 East Blackland Road, Temple, TX 76502.

Beven, K., Kirkby, MJ. (1979): A physically based variable contributing area model of basin hydrology Hydrological Science Bulletin, 24, pp. 43-69

Bronstert, A., Plate, E.J. (1997): Modelling of Runoff Generation and Soil Moisture Dynamics for Hillslopes and Micro-Catchments. Journal of Hydrology, 198 (1-4), 177-195.

Faeh, A.O., Scherrer, S., Naef, F. (1997): A combined field and numerical approach to investigate flow processes in natural macroporous soils under extreme precipitation Hydrol. Earth Syst. Sci., 1, 787–800, 1997

Krysanova, V., Müller-Wohlfeil, D.-I., Becker, A. (1998): Development and test of a spatially distributed hydrological/water quality model for mesoscale watersheds. Ecological Modelling, 106, 261-289.

Liang, X., Lettenmaier, D. P., Wood, E. F., and Burges, S. J. (1994): A simple hydrologically based model of land surface water and energy fluxes for general circulation models, J. Geophys. Res., 99, 675.

Ludwig, K., Bremicker, M. (Eds) (2006): The Water Balance Model LARSIM - Design, Content and Applications, Freiburger Schriften zur Hydrologie, No. 22

Michel, C. (1983): Que peut-on faire en hydrologie avec modèle conceptuel à un seul paramètre?, La Houille Blanche, 69, 39–44.

Nielsen, S. A. and Hansen, E. (1973): Numerical simulation of the rainfall-runoff process on a daily basis, Nord. Hydrol., 4, 171–190.

Smith, RE., Goodrich, DC., Woolhiser, DA., Unkrich., CL. (1995): KINEROS – A kinematic runoff and erosion model; Chapter 20 in V.P. Singh (editor), Computer Models of Watershed Hydrology, Water Resources Publications, Highlands Ranch, Colorado, 1130 pp.

Todini, E. (1996): The ARNO rainfall-runoff model, J. Hydrol., 175, 339–382.

Zehe, E., Maurer, T., Ihringer, J., Plate, E. (2001): Modeling water flow and mass transport in a loess catchment, Physics and Chemistry of the Earth, Part B: Hydrology, Oceans and Atmosphere, Volume 26, Issues 7–, 487-507.

Zhao, R.J. (1977): Flood forecasting method for humid regions of China. East China Institute of Hydraulic Engineering, Nanjing, China.

Zhao, R.J., Zhang, Y.-L., Fang, L.-R., Liu, X.-R. & Zhang, Q.-S. (1980): The Xinanjiang model. In: Hydrological Forecasting (Proc. Oxford Symposium, April 1980), pp 351-356. IAHS Publ. 129, IAHS Press, Wallingford, UK.

---

## Referee Comment (RC4)

*Review of the paper*
**A retrospective on hydrological modelling based on half a century with the HBV model**
*by Jan Seibert and Sten Bergström*
*submitted for publication in*
*HESS*

This is a nicely written story about the HBV model in the context of hydrological model developments in the last 50 years. The style of the paper is somewhat unusual - casual but certainly acceptable. While there are a large number of facts in the paper, it also contains subjective opinions which may lead to discussions. In general the paper is acceptable my remarks may be considered by the authors in case they revise their manuscript.

1. While the story of the model and the underlying concepts are discussed in detail, the model itself is not described. Including the basic model equations would not increase the length of the paper too much, but would allow the student readers who may not know the model to better understand the paper.

2. The discussion on physically based modelling is somewhat simplified. For me mass conservation is not the only physical basis what one has in describing the hydrological cycle. The argument that SHE needs much more parameters then HBV is valid, but SHE produces more interpretable outputs - which could be verified and used for some calibration. Further the *physically based* models usually have better control of the internal state variables. In my view HBV is a rainfall/runoff model with all its advantages and disadvantages. Physically based models may be used for other hydrological questions too due to their detailed process descriptions.

3. In the discussion of the processes evapotranspiration is completely missing. You could add a few sentences for the sake of completeness.

4. Model quality does not only depend on the available data but on the quality of the data and the variability within the catchment. HBV is relatively random error tolerant (not bias tolerant) - which is an advantage additional to ppp.

5. We used the HBV for the assessment of land use change using simultaneous calibration of model parameters based on catchment properties (Hundecha and Bardossy 2004). In general regional calibration is an option for simple models like HBV to use space for time and gain valuable additional information.

6. What do you mean by insignificant subroutines ? Shouldn't it be insignificant process details?

7. In my opinion one of the interesting questions of the future is that to what extent can simple models such as HBV cope with the present and continuously increasing amount of (often very noisy) indirect information.

Reference:

Hundecha, Y. and Bárdossy, A.: Modeling of the effect of land use changes on the runoff generation of a river basin through parameter regionalization of a watershed model, *Journal of Hydrology*, **292**, 281—295, 2004

---

## Author Comment (AC3)

We appreciate this detailed review of our manuscript. We are glad that our somewhat unusual style is seen as positive. The reviewer lists three more general remarks and a number of detailed comments. Below we discuss how we will address these in the revised version of the manuscript (reviewer comments in blue italic font)

*"Physically based" vs. "conceptual" ?*

This is an interesting question. We agree that there is no clear distinction which makes some models physically-based and others not. However, as discussed in the review of Jens Christian Refsgaard, there is a difference in the degree to which models are physically-based. We fully agree that in the end, the crucial question is, which situations/tasks a model is suitable for. We will clarify this in the text.

*Number of parameters and desire for optimization*

There is a difference between how many parameters are in a model and how many are actually allowed to take different values. As discussed in the review, this (also) depends on how much spatial variability is considered. In the revision, we will follow the reviewer's advice and extend the discussion on when a distributed model is (not) needed.

*Need for a classification of hydrological models*

As also noted in the review of of Jens Christian Refsgaard, we used the term hydrological models in a rather narrow sense and will clarify that we in this manuscript are focusing on catchment (runoff) models. We agree that also within these models there are different 'families' and will extend the text slightly to clarify this.

*- Line 115: "advocation for realism in model development"? What is 'realism' in this regard? And Is this still valid for today?*

We will change the word 'realism' to 'best practices'

*- Line 130: Instead "represent all hydrological processes", better term it "represent all relevant hydrological processes",*

Agreed, will be changed

*- Line 285: what do you mean with "re-organising observed flood generating factors'?*

We will change 're-organising observed flood generating factors' to 'combining observed flood generating factors such as heavy rainfall, extreme snowmelt situations and wet antecedent conditions'

*- Line 300: "risk for overparameterisation"? You may discuss it in the relation with complexity of model and nature? Also nature may have several means to come to one state? What can one do in this regards?*

We will add 'i.e., the inability to determine one single best parameter set' for clarification

*- Line 315: "groundwater dynamics"? You may elaborate a bit how far the catchments wetness in the HBV model can be related to observed gw-dynamics.*

Groundwater dynamics were adressed in some early applications of a modified version of the HBV-model as described by Bergström and Sandberg (1983) and we have used groundwater dynamics as

additional criteria in previous studies (e.g., Seibert, 2000, HESS). We will extend section 4.4 and discuss opportunities and challenges of multi-criteria model evaluations.

*Seibert, J., 2000. Multi-criteria calibration of a conceptual rainfall-runoff model using a genetic algorithm. Hydrology and Earth System Sciences, 4(2): 215-224.*

*Bergström, S. and Sandberg, G. (1983) Simulation of groundwater response by conceptual models - Three case studies. Nordic Hydrology, Vol. 14, No. 2.*

*- Lines 465-480: about the application for ungauged catchments: You mention the option of ensemble applications (many sets of parameters). You may elaborate a bit more on this. For my understanding this is a very valuable approach.*

An early indication of the value of using ensembles for HBV model simulations can be found in Seibert and Beven (2009, HESS) where we found that the ensemble mean performed generally better than the best single parameter set (see Figure 2). This finding was more recently confirmed in studies such as the two we refer to here (Seibert et al., 2018; Pool et al., 2021). We certainly add a bit more text on this approach.

*- Figure 6: The value of a model depends on its purpose!! If the purpose is to look like the reality, the left figure is much "better".*

We agree, but the question we posed is 'Which modell will fly?' – and trying to let the left plane fly will not end well ☺

Again, thanks for these valuable comments and interesting thoughts that will help us clarify the manuscript.

Best regards,

Jan and Sten

---

## Author Response (AR1)

**Comments by Keith Beven**

Thanks for handling our manuscript. We are especially thankful for getting these great colleagues as reviewers.

> *1. Line 450 and Section4.4 - you give no references to use of HBV with uncertainty estimation*

Added

> *2. L500 Wide use does not imply that parameter variations between regions or catchments are small, and seems immediately in conflict with Line505.*

We agree and reformulated this section.

**Comments by Jens-Christian Refsgaard**

Thanks for these valuable comments on our manuscript. The reviewer is right that we, in this manuscript for the special issue on History in Hydrology, took a somewhat inward look by focusing on the history of the HBV model. A full review of the history of catchment modelling would be beyond the scope of this manuscript, but we will try to be a bit more self-reflecting in the revised version.

Beyond we shortly discuss we addressed the points raised in the reviewers list:

1. Yes, the term hydrological modeling is broader than we use it here. Your suggestion was to change this to rainfall-runoff modelling. As snow is an important part in many HBV studies, this should be precipitation-runoff modelling, or perhaps even precipitation-temperature-runoff modelling – but then there is also potential evapotranspiration as input. Therefore, we changed this to 'catchment modelling' and clarified at several places that the focus of models such as HBV is catchment runoff.
2. Yes, we agree that for other types of hydrological models other issues than those discussed here from the aspect of (conceptual) catchment models will be important. We clarifed the more narrow focus of the discussion in our manuscript (see also point 1).
3. Good point, we did not want repeat the full (interesting) discussion between these papers, but now refer to this discussion.
4. Yes, we realize that our use of the term hydrological modelling might cause confusion as we are focusing on catchment (runoff) models here (and not all the other types of hydrological models), we clarified this throughout the manuscript
5. Thanks
6. Agreed, we changed the heading to 'Early catchment models'
7. We here wanted to express that the differences between a simple model and a model based on equations such as Darcy might not be so large if the latter is applied using elements or parameterizations that do not capture the real-world heterogeneities. We clarified this section.
8. Good point; we implemented these thoughts in the revised version.
9. We agree and clarified these important details of Breuer et al. (2009) in the revised version.
10. We agree that many hydrologists working with modelling would say that all kinds of catchment models need some form of calibration. However, we also still meet the notion that more complex models with 'measurable' parameters would not require calibration or at least not as much calibration (thus, using other terms such as tuning) by colleagues working with these, in their eyes superior, models. We appreciate the nuanced discussion in

Hrachowitz & Clark (2017) and also the reviewer statement that the view on the need for calibration has changed over the years. We included the H&C reference and also added a pers.com. reference to the reviewer to include the changing-ideas-on-calibration point.

11. The (international) success of the HBV model is in our opinion much related to the 3 P-s (parsimony, performance, persistence) as mentioned in the introduction and above in section 3. The parsimony made the model code easy to understand (and re-program), the HBV model often performed well especially in comparison studies, and it certainly helped that the model was intensely used by SMHI and the hydrological services of Norway and Finland, and became a standard tool for the Nordic hydropower industry (opposite to a single researcher or research group as was the case for other models). We added these thoughts in the beginning of section 3.

12. We agree that new sources of data (including spatial patterns derived from remote sensing) might open new opportunities, but as we write there is a challenging balance between more available data and more model complexity to make these data directly usable for model evaluation. We argue that this applies to any type of catchment model but also see some promising studies such as the one by Stisen et al.. We added text in section 4.4 to clarify that we see multi-criteria calibration as a suitable way forward although this approach is challenging in practice.

**Review Axel Bronstert**

We appreciate this detailed review of our manuscript. We are glad that our somewhat unusual style is seen as positive. The reviewer lists three more general remarks and a number of detailed comments. Below we discuss how we will address these in the revised version of the manuscript (reviewer comments in blue italic font)

*"Physically based" vs. "conceptual" ?*

This is an interesting question. We agree that there is no clear distinction which makes some models physically-based and others not. However, as discussed in the review of Jens Christian Refsgaard, there is a difference in the degree to which models are physically-based. We fully agree that in the end, the crucial question is, which situations/tasks a model is suitable for. We clarified this in the text.

*Number of parameters and desire for optimization*

There is a difference between how many parameters are in a model and how many are actually allowed to take different values. As discussed in the review, this (also) depends on how much spatial variability is considered. We added some text on the discussion of when to use which type of model.

*Need for a classification of hydrological models*

As also noted in the review of Jens Christian Refsgaard, we used the term hydrological models in a rather narrow sense and clarified that we in this manuscript are focusing on catchment (runoff) models. We agree that also within these models there are different 'families' and extended the text on model classification.

*- Line 115: "advocation for realism in model development"? What is 'realism' in this regard? And Is this still valid for today?*

We changed the word 'realism' to 'best practices'

*- Line 130: Instead "represent all hydrological processes", better term it "represent all relevant hydrological processes",*

Agreed, changed

*- Line 285: what do you mean with "re-organising observed flood generating factors'?*

We changed 're-organising observed flood generating factors' to 'combining observed flood generating factors such as heavy rainfall, extreme snowmelt situations and wet antecedent conditions'

*- Line 300: "risk for overparameterisation"? You may discuss it in the relation with complexity of model and nature? Also nature may have several means to come to one state? What can one do in this regards?*

We added 'i.e., the inability to determine one single best parameter set' for clarification

*- Line 315: "groundwater dynamics"? You may elaborate a bit how far the catchments wetness in the HBV model can be related to observed gw-dynamics.*

Groundwater dynamics were adressed in some early applications of a modified version of the HBV-model as described by Bergström and Sandberg (1983) and we have used groundwater dynamics as additional criteria in previous studies (e.g., Seibert, 2000, HESS). We added text in sections 3.1 and 4.4 and discuss opportunities and challenges of multi-criteria model evaluations.

*- Lines 465-480: about the application for ungauged catchments: You mention the option of ensemble applications (many sets of parameters). You may elaborate a bit more on this. For my understanding this is a very valuable approach.*

An early indication of the value of using ensembles for HBV model simulations can be found in Seibert and Beven (2009, HESS) where we found that the ensemble mean performed generally better than the best single parameter set (see Figure 2). This finding was more recently confirmed in studies such as the two we refer to here (Seibert et al., 2018; Pool et al., 2021). We added a few words and references on this approach.

*- Figure 6: The value of a model depends on its purpose!! If the purpose is to look like the reality, the left figure is much "better".*

We agree, but the question we posed is 'Which model will fly?' – and trying to let the left plane fly will not end well ☺

**Comment Stein Beldring**

We fully agree that many HBV concepts can also be used in distributed approaches. We appreciate the reminder us about these papers which we added as a good example on a distributed version of the HBV concept.

**Comments by András Bárdossy**

We appreciate the positive assessment of our manuscript and the helpful list of comments. Especially we like the statement, "This is a nicely written story about the HBV model in the context of hydrological model developments in the last 50 years. The style of the paper is somewhat unusual - casual but certainly acceptable." Here we reply to the seven points raised by the reviewer:

1) Good point. We are afraid to make the manuscript harder to read if we insert the model equations. However, we added a reference to Seibert&Vis (2012) and could also add an appendix similar to the one in Seibert (1999).

2) We agree on this potential for physically-based models like SHE, although we might be less optimistic that the additional constraints (always) compensate for the increased degrees of freedom. But of course, there are applications where we need a better representation of internal fluxes and states. We clarified this in the revised version.

3) Thanks for making us aware that we missed discussing evapotranspiration in any detail. We added some information on evapotranspiration, especially how to derive Epot for the input, in section 3.1.2.

4) The often limited sensitivity to random errors is indeed an advantage of HBV, but also other water balance accounting errors. We agree that this is another reason why such models often perform well despite not-so-good input data. We see this related to one of the three P-s and added this valuable thought to the discussion of model performance.

5) Thanks for making us aware of this interesting study. We added the use of HBV for land-use change studies as suggested by Hundecha and Bárdossy (2004) in section 4.1

6) With "insignificant subroutines" we meant subroutines that are not affecting the (runoff) simulation substantially. Following the comment, we changed this to "subroutines representing insignificant process details"

7) We agree that it is an interesting question how models like HBV can make use out of different types of data and tried to make this clearer in the revision.

---

## Author Response (AR2)

Dear Keith,

Thanks for your comments that helped us to further clarify the text. Below our replies to your comments (in blue).

Best regards,

Jan and Sten

Row 11. "1960s??.Stanford / Dawdy & O'Donnell? "

Comment Sten: "Jag minns vagt artikeln av Dawdy och O'Donnell, men den hade inte lika stor betydelse som Nashs artiklar vid utvecklingen av HBV."

We added 1960s in the abstract. (we also added the reference to Dawdy and O'Donnell, see below)

Row 30. Keith: "fix or fit or program?"

construct

Row 36. Keith: "add. Around 2000. Model predictions are necessarily uncertain - how can we estimate that uncertainty given the data and computer resource available?"

Good point, we added this (in some modified words)

Row 110 och 178. We realized that we had used "Early catchment models" for both 2.1 and 2.2. and changed this

Row 118. Keith: "e.g. Dawdy and O'Donnell already in 1965".

Thanks for this good suggestion, added

Row 199. Keith: "though my 1989 paper already pointed out that physically-based models were necessarily conceptual approximations if using effective parameter values at the grid scale since Richards does not average linearly (even if it applies at all)".

Absolutely, we added some text and ref to your 1989 paper

Row 217-218. Keith: " This is ignoring the commensurability issue - you cannot match observed and predicted internal states????"

We agree that comparisons are tricky but still see value in them. We modified and extended the text to make this clearer.

Row 352. Keith: "Though [as an aside] of course they were all pretty bad since even a albedo parameters needed to vary significant; y from year to year".

Hmm, but we guess this is another discussion, probably not worth going into more detail and critic here.

Row 487. Keith: "Would be perhaps interesting to add briefly how many subcatchments were modelled and how parameters were estimated for rivers with and without adequate discharge estimates"

We modified and extended the text.

Row 505: Keith: "there are different forms of regional calibration - what do you actually do?"

This refers to the approach used by Hundecha and Bárdossy. We modified the text to clarify this.

Row 522: Keith: "Though it would be nice to have some results to demonstrate here!!!"

This is an aspect we are currently looking at in more detail in Zurich as a follow up of Sandra Pool's work. We have some preliminary results but presenting these here would require explain a lot of methods etc. However, we plan to submit a manuscript on this in the (hopefully near) future.

Row 539 och 558: Keith: " But were the land surface parameterisations in the climate models consistent with the HBV representation of the hydrology????"

We modified the text to clarify the inconsistency in the modelling process.

[revised manuscript text omitted]